# Percutaneous Radiology Gastrostomy (PRG)-Associated Complications at a Tertiary Hospital over the Last 25 Years

**DOI:** 10.3390/nu14224838

**Published:** 2022-11-15

**Authors:** Ana Piñar-Gutiérrez, Pilar Serrano-Aguayo, Silvia García-Rey, Rocío Vázquez-Gutiérrez, Irene González-Navarro, Dolores Tatay-Domínguez, Pilar Garrancho-Domínguez, Pablo J. Remón-Ruiz, Antonio J. Martínez-Ortega, Verónica Nacarino-Mejías, Álvaro Iglesias-López, José Luis Pereira-Cunill, Pedro Pablo García-Luna

**Affiliations:** 1UGC Endocrinología y Nutrición, Hospital Universitario Virgen del Rocío, 41013 Seville, Spain; 2Servicio de Radiología, Unidad de Radiología Intervencionista, Hospital Universitario Virgen del Rocío, 41013 Seville, Spain

**Keywords:** gastrostomy, enteral nutrition, complications, fluoroscopy, interventional radiology

## Abstract

Objectives: We aimed to describe and compare the complications associated with different percutaneous radiologic gastrostomy (PRG) techniques. Methods: A retrospective and prospective observational study was conducted. Patients who underwent a PRG between 1995–2020 were included. Techniques: A pigtail catheter was used until 2003, a balloon catheter without pexy was used between 2003–2009 and a balloon catheter with gastropexy was used between 2015–2021. For the comparison of proportions, X^2^ tests or Fisher’s test were used when necessary. Univariate analysis was performed to study the risk factors for PRG-associated complications. Results: *n* = 330 (pigtail = 114, balloon-type without pexy = 28, balloon-type with pexy = 188). The most frequent indication was head and neck cancer. The number of patients with complications was 44 (38.5%), 11 (39.2%) and 54 (28,7%), respectively. There were seven (25%) cases of peritonitis in the balloon-type without-pexy group and 1 (0.5%) in the balloon-type with-pexy group, the latter being the only patient who died in the total number of patients (0.3%). Two (1%) patients of the balloon-type with-pexy group presented with gastrocolic fistula. The rest of the complications were minor. Conclusions: The most frequent complications associated with the administration of enteral nutrition through PRG were minor and the implementation of the balloon-type technique with pexy has led to a decrease in them.

## 1. Introduction

In 1980, Gauderer and Ponsky described percutaneous endoscopic gastrostomy (PEG) [1] as an alternative method to gastrostomy by laparotomy. Since then, various endoscopic gastrostomy techniques, percutaneous radiological gastrostomy (PRG) and laparoscopic gastrostomies have been developed. Currently, surgical gastrostomies have been restricted to patients for whom endoscopic and radiological techniques are not possible [2] because it is more invasive, requires more anesthesia and is more expensive. Endoscopic and radiological techniques are therefore the most widespread, and their availability has increased in recent years in hospitals around the world [3]. These can be performed using the PUSH technique, which consists of pushing the probe through the guidewire until the tip of the dilator appears through the abdominal wall and stretching from this point, or by the PULL technique, in which a puncture is made through the abdominal wall and then a traction is performed, dragging the guidewire through the stomach, esophagus, pharynx and mouth.

Studies describing and comparing the results of both techniques in terms of the associated complications have presented disparate results. However, in general, they have shown similar outcomes with respect to safety [4,5]. PRG has also been shown to have a high success rate, even in patients in whom PEG was previously attempted [6,7], and in those with head and neck cancer and/or esophagogastric pathology, in whom endoscopic passage is not possible [8]. It also has the advantage of lower costs [9], the need for less deep sedation (especially important in patients with a higher risk of aspiration) and the fact that it does not routinely require prophylactic antibiotherapy [5]. Despite all this, PEG is still the most frequently used technique in most centers [8,10]. Moreover, in many hospitals, PGR has not yet been implemented, leaving surgical gastrostomy as the only alternative when PEG is not technically possible or not available.

Although gastrostomies can be performed with the aim of decompressing the stomach, their most frequent indication is enteral nutritional support because this route is recommended over the nasogastric tube in patients who require this support for a period of more than 4–6 weeks [11]. Although these procedures have a low morbimortality, their performance is not free of complications [12,13]. Among these complications are minor ones (more frequent), such as exudate, irritation, granuloma, tube obstruction, tube leakage, etc., and other major ones (less frequent, but can be lethal), such as bronchoaspiration, peritonitis, gastric perforation or colonic perforation in cases of cologastric fistulas and massive bleeding. That is why there is a constant search for new techniques and improvements in them to reduce complications. An example of this is the use of gastropexy in PRGs, which has made it possible to reduce the risk of peritonitis [5] or the abandonment of pigtail tubes for safer ones [14].

Based on these advances and on the results of complications obtained with the PRG techniques used in our center [15], these techniques have been replaced by others, with three different techniques having been performed in the last 25 years. Our center is a tertiary hospital with a nutrition unit, specializing in the care of these patients, and an interventional radiology unit with extensive experience in these procedures.

The primary objective of our study is to describe the complications associated with the different PRG techniques performed in the last 25 years at our center. The secondary objectives were to compare the complications between the currently performed technique and the previously performed techniques, to describe the indications for these techniques and their evolution during this period, and to search for risk factors for PRG-associated complications.

## 2. Materials and Methods

### 2.1. Study Design

A retrospective observational study was conducted in which all adult patients who had undergone a PRG at the Virgen del Rocío University Hospital and who were subsequently followed up at the hospital’s nutrition unit between 1995 and 2020 were included.

### 2.2. Data Collection, Variables and Follow-Up

Data collection was performed retrospectively until 2008, before subsequently being prospectively performed using the clinical records of the Andalusian Health Service. Written informed consent was obtained from all the patients before the procedure. All the required clinical and ethical guidelines of our center were followed.

The study variables included were as follows: sex; age; date of performance; indication for gastrostomy (head and neck tumors, esophageal tumors, non-tumor esophageal diseases, ALS, other neurological diseases, severe malabsorption, maxillofacial diseases and others); type of PRG; presence of complications; presence of major complications (peritonitis, need for invasive mechanical ventilation (IMV) due to respiratory distress after the procedure and gastrocolic fistula); presence of minor complications (exudate, irritation (reddening of the skin around the stomach), balloon leakage, obstruction of the tube lumen, stoma dilatation, bleeding not requiring transfusion, granuloma, balloon rupture and/or local infection (inflammation and purulent leakage with the presence of microorganisms in the culture and the need for antibiotic treatment)); and exitus, secondary to a PRG-associated complication. The follow-up period was carried out until the patient’s loss of follow-up in the nutrition unit, which took place between 3 and 24 months.

### 2.3. Statistical Analysis

Statistical analysis was carried out using Statistical Package for Social Science (SPSS^®^) version 25 for Windows (IBM Corporation, New York, NY, USA). Descriptive analysis was performed by obtaining the median and the quartiles of the quantitative variables (expressed as P50 (P25–P75)), as well as the frequency of the qualitative variables (expressed as n (%)). For the comparison of proportions, X2 tests or Fisher’s test were used when necessary. Univariate analysis was performed to study the existence of risk factors associated with PRG-associated complications in the total sample. To avoid confounding factors, we stratified a specific PRG technique by performing a multivariate analysis with each risk factor studied. A *p* value < 0.05 was considered statistically significant.

### 2.4. Techniques

PRGs were performed in all cases by radiology specialists with extensive experience in interventional radiology, using radioscopic assistance with fluoroscopic guidance. A previous CT scan to rule out the interposition of viscera between the gastric chamber and the abdominal wall was mandatory as part of our hospital’s protocol for GPR placement.

Until September 2003, only pigtail probes (front-facing type) were used in our center. Subsequently, simple balloon probes were used until 2009, without gastropexy (hereafter and for simplicity, this technique will be referred to as balloon-type without pexy). Finally, from 2009 to 2020, the PUSH technique of internal balloon support and an additional anchoring system with pexy was used (hereafter and for simplicity, this technique will be referred to as balloon-type with pexy). In all of them, 12–18 French caliber tubes were used, and antibiotic prophylaxis was not used. Enteral nutrition began 6 h after the procedure. After admission, the patients were assessed on an outpatient basis by nurses or physicians specializing in clinical nutrition at one month, and then every three months.

## 3. Results

A total of 330 patients were included. A total of 114 patients underwent a PRG with a pigtail tube between 1995 and 2003, 28 patients underwent a balloon-type PRG without pexy between 2003 and 2009, and 188 patients underwent a balloon-type PRG with pexy between 2009 and 2020.

The demographic characteristics (age and sex) of the included patients are shown in Table 1.

The pathologies that motivated the indication of PRG in these patients are shown in Table 2.

The complications associated with the different PRG techniques performed in the period from 1995 to 2020 are shown in Table 3. 

One patient died in 2019 after a balloon-type PRG with pexy in the context of peritonitis associated with the procedure, in addition to severe upper-airway obstruction due to a well-differentiated pharyngeal carcinoma cT4N3Mx.

As for gastrocolic fistula, there were two cases (1%) associated with the balloon-type PRG with-pexy technique. The first case was a 73-year-old man with a head and neck tumor who, 10 months after the procedure, presented with diarrhea, similar to the enteral nutrition formula he was receiving, as well as severe malnutrition. The gastrostomy had to be surgically removed, requiring a resection of part of the colon. Subsequently, a new surgical gastrostomy was performed using the Janeway technique. The second patient was a 64-year-old man with a head and neck tumor who, after tube replacement, presented with diarrhea and polymicrobial bacteremia (*S. aureus*, *K. aerogenes* and *E. faecalis*), secondary to the fistula created by the gastrostomy tube and, as a consequence, presented with severe malnutrition. He was treated with intravenous antibiotherapy (cefazolin and piperacillin/tazobactam) and a laparoscopy-assisted percutaneous gastrostomy (PLAG [16]) was performed.

When comparing only the pigtail PRG with the balloon-type PRG, with pexy there was a statistically significant difference in total complications (38.5% vs. 28.7%, *p* = 0.05), exudate (26.3% vs. 14.3%, *p* = 0.008) and local infection (0% vs. 4.7%, *p* = 0.013).

The results of the search for risk factors for suffering any complication associated with PRG, both in the total sample and in each of the techniques performed, are shown in Table 4.

## 4. Discussion

This study evaluates the rate of complications associated with the performance of 330 PRGs in adult patients with different techniques employed over the last 25 years at a tertiary hospital: pigtail; balloon-type without pexy; and balloon-type with pexy, resulting in an associated 38.5%, 39.2% and 28.7% of complications, respectively. Most of these were minor, with the most frequent being exudate. However, the use of the balloon tube without pexy was associated with a peritonitis rate of 25%; therefore, it was abandoned early when the problem became known (and this is the reason why it was only used in 28 patients in our data). The use of pexy has been an innovation that makes the technique safer by adding an extra fixation of the stomach to the abdominal wall until healing of the gastrostomy is complete. Without pexy, the only anchoring mechanism is the balloon, which is susceptible to deflation or rupture, leaving the stomach with a free orifice in the abdominal cavity. Mortality due to complications of the technique itself was less than 1%.

In 1981, Preshaw described for the first time the performance of percutaneous gastrostomies using radiological techniques [17]. At present, although there are different variants, the most widely used technique is that of Seldinger, in which the stomach is fixed to the abdominal wall with T-fasteners. Subsequently, a trocar is introduced into the gastric chamber through a metallic guidewire and a dilation system on the guidewire, dilating the orifice to a diameter similar to that of the tube to be used. As a previous step, it is necessary to insufflate the stomach, which can be conducted through a fine caliber nasogastric tube or even an endovascular catheter when there is marked stenosis of the upper digestive tract. As previously mentioned, the use of new tubes and the routine performance of gastropexy decreases the associated complications [5,14]. In addition, new PULL techniques are being developed that may increase the success rate and reduce complications, although more studies are needed in this regard [7,18,19].

At present, the performance of PRGs is less widespread than that of PEGs. This may be due to the fact that, in most centers, there is greater experience with the latter, as well as a greater availability of endoscopists than interventional radiologists. When analyzing the results currently published regarding the differences in complications between the two techniques, we found studies in which there were no statistically significant differences [20,21,22,23], others in which PRGs presented a greater number of complications [8,24] and vice versa [25,26]. However, the available literature is often limited to small observational, retrospective, single-center studies that vary in their indications. Some studies found differences in specific complications. For example, in the study by Vidhya et al. [27] only the proportion of tube leakage was more frequent with the radiological technique. What is also interesting in this regard is the study by Cherian et al. [26], in which both tube leakage and obstruction were more frequent in PRGs. Finally, of note is the study by Kohli et al. [4], performed with a national North American database of readmissions using coded diagnoses, in which infections, colon perforation and hemorrhages requiring a blood transfusion were more frequent in PRGs. None of these complications exceeded 10% in our study, and the most severe ones (colonic perforation due to gastrocolic fistula and bleeding) were extremely infrequent (1% and 0.5%, respectively) in the technique currently used in our center. When comparing both techniques, it is also important to highlight that a PRG requires less deep sedation and does not require prophylactic antibiotherapy [5]. Furthermore, it is less expensive [9] and is useful in cases in which a PEG has not been possible. In most cases, this is due to the difficulty in passing the endoscope because of tumor stenosis [6,7,8]. Therefore, we believe that it is important that clinical nutrition units with extensive experience with PRGs, such as ours, publish data on complications, as this may encourage other centers to implement or increase the availability of this technique. This could benefit patients as it would constitute an alternative to PEGs in cases in which it could not be performed and would avoid the performance of surgical techniques in patients in whom a PRG would be feasible.

In our study, the data on peritonitis associated with the substitution of the pigtail catheter technique for the balloon-type without-pexy technique (25% of patients (7 out of 28)) is noteworthy. For this reason, this technique was abandoned early, and a very low number of gastrostomies were performed until the gastropexy technique was implemented, as a result of which the percentage of peritonitis was reduced to 0.5% (1 patient out of 188).

Regarding other major complications, only two patients of the total sample presented with a gastrocolic fistula. This data is similar to that reported by other authors [28]. Although it is an infrequent complication, in order to diagnose it, it is necessary to consider it in the differential diagnosis of a patient with a gastrostomy who presents, after tube replacement, with postprandial diarrhea with characteristics similar to the nutritional supplements that are being given through the tube and, as a consequence, presents with malnutrition in spite of adequate therapeutic adherence [29,30].

If the 28 patients who underwent a balloon-type PRG without pexy are omitted, comparing the pigtail balloon-type PRG with the gastropexy PRG shows a significant reduction of almost 10% in total complications, as described in the literature [5].

Regarding indications, it is worth noting the increase in the indication for a PRG for ALS in the last period (2009–2020). This is due to the implementation of a specific multidisciplinary unit in which clinical nutrition specialists participate and which has led to an improvement in the care of these patients; therefore, it has led to a greater indication for the percutaneous gastrostomy technique that requires shallower sedation. In addition, in our center, a PRG is usually performed before a PEG in these patients due to the former’s lesser use of sedation and the consequent lower risk of respiratory difficulties, an important cause of mortality in people with ALS. This daily clinical practice is further supported by the results obtained in our study whereby ALS was associated with a lower risk of total complications.

Finally, the only risk factor associated with total complications was esophagogastric cancer as the pathology that led to the indication for gastrostomy. Therefore, it is necessary to evaluate this subgroup of patients in the future in order to find which factors were associated with a higher percentage of complications and whether other gastrostomy techniques may actually be safer for them.

This study is limited in the first place by its observational design and retrospective data collection. It would have been interesting to take into account variables that were not collected and that can affect the number of complications, such as the use of anticoagulants, diameter of the gastrostomy tubes, duration of the procedures, follow-up time, previous nutritional status of the patient, nutritional formulas used and comorbidities of the patients. Such a collection was not possible because the oldest records are not in a digital format, and these data were not present in the records. 

The inclusion in the study of patients with different pathologies and indications may represent another bias, and subanalyses should be performed in the future in these groups because complications could vary among them and the indication for one technique or another could depend on this. Additionally, the choice of technique should be made as individually as possible. Finally, the results of our study may not be extrapolated to other hospitals because ours is a tertiary-level hospital with multidisciplinary units where less frequent and more complex pathologies are treated, and the indications may differ with respect to other centers.

As for strengths, we believe that the high number of PRGs and the long period evaluated is a factor to be taken into account, as well as the fact that all the PRGs were indicated and followed by the same group of professionals from the nutrition unit and interventional radiology unit. Logically, there were with new incorporations and absences during the 25 years studied, but under the same action protocols.

## 5. Conclusions

We present a series of 330 patients who underwent PRGs in a tertiary hospital in the last 25 years. The indication for a PRG has varied over this period, the most frequent being head and neck cancer, followed by ALS, at present. The most frequent PRG-associated complications were minor, and the implementation of the new balloon-type technique with pexy led to a decrease in their occurrence. The balloon-type technique without pexy is not recommended and had to be abandoned in our center due to a high percentage of associated peritonitis. Further studies in centers in different regions and at different levels will allow us to increase our knowledge of these techniques in order to develop improvements and reduce the risk of associated complications.

## Figures and Tables

**Table 1 nutrients-14-04838-t001:** Sex and age distribution of patients undergoing PRG at the Hospital Universitario Virgen del Rocío between 1995 and 2020.

	Pigtail PRG (1995–2003, *n* = 114)	Balloon-Type PRG without Pexy (2003–2009, *n* = 28)	Balloon-Type PRG with Pexy (2009–2020, *n* = 188)
Male sex	88 (77.2%)	15 (53.6%)	126 (67%)
Age (years)	52 (40–62)	63 (52–74)	62 (57–70)

**Table 2 nutrients-14-04838-t002:** Indication of PRG in 330 patients seen in the nutrition unit of the Virgen del Rocío University Hospital between 1995 and 2020.

	Pigtail PRG (1995–2003, *n* = 114)	Balloon-Type PRG without Pexy (2003–2009, *n* = 28)	Balloon-Type PRG with Pexy (2009–2020, *n* = 188)
Head and neck cancer	51 (44.7%)	16 (57.1%)	73 (38.8%)
Esophagogastric cancer	13 (11.4%)	3 (10.7%)	21 (11.1%)
Non-tumoral esophagogastric pathology	0 (0%)	0 (0%)	3 (1.5%)
ALS	2 (1.7%)	6 (21.4%)	58 (30.8%)
Neurological diseases	42 (36.8%)	1 (3.5%)	26 (13.8%)
Severe malabsorption	0 (0%)	0 (0%)	1 (0.5%)
Maxillofacial pathology	3 (2.6%)	0 (0%)	1 (0.5%)
Others	3 (2.6%)	0 (0%)	9 (4.7%)

**Table 3 nutrients-14-04838-t003:** Complications associated with PRG in 330 patients treated at the nutrition unit of the Hospital Universitario Virgen del Rocío between 1995 and 2020.

	Pigtail PRG (1995–2003, *n* = 114)	Balloon-Type PRG without Pexy (2003–2009, *n* = 28)	Balloon-Type PRG with Pexy (2009–2020, *n* = 188)	*p*
Patients with any complication	44 (38.5%)	11 (39.2%)	54 (28.7%)	0.16 ^a^
Exitus due to PRG-related complications	0 (0%)	0 (0%)	1 (0.5%)	1 ^b^
Peritonitis	0 (0%)	7 (25%)	1 (0.5%)	<0.001 ^b^
Need for IMV due to respiratory distress after procedure	0 (0%)	0 (0%)	0 (0%)	
Gastrocolic fistula	0 (0%)	0 (0%)	2 (1%)	0.894 ^b^
Exudate	30 (26.3%)	2 (7.1%)	27 (14.3%)	0.01 ^a^
Irritation	0 (0%)	0 (0%)	0 (0%)	
Leakage	8 (7%)	2 (7.1%)	12 (6.3%)	0.972 ^a^
Obstruction	2 (1.7%)	2 (7.1%)	9 (4.7%)	0.203 ^b^
Stoma dilation	0 (0%)	1 (3.5%)	3 (1.5%)	0.128 ^b^
Bleeding	0 (0%)	1 (3.5%)	1 (0.05)	0.163 ^b^
Granuloma	17 (14.9%)	1 (3.5%)	22 (11.7%)	0.248 ^a^
Breakage	2 (1.7%)	0 (0%)	5 (2.6%)	0.847 ^b^
Local infection	0 (0%)	1 (3.5%)	9 (4.7%)	0.031 ^b^

^a^ X^2^ Test; ^b^ Fisher test.

**Table 4 nutrients-14-04838-t004:** Risk factors for total complications after PRG in 330 patients treated at the nutrition unit of the Hospital Universitario Virgen del Rocío between 1995 and 2020.

	OR (95% CI); *p*			
	Pooled (1995–2020, *n* = 330)	Pigtail PRG (1995–2003, *n* = 114)	Balloon-Type PRG without Pexy (2003–2009, *n* = 28)	Balloon-Type PRG with Pexy (2009–2020, *n* = 188)
Sex (woman)	1.19 (0.738–1.92); 0.476	1.253 (0.772–2.035); 0.362	1.3 (0.64–2.651); 0.465	1.23 (0.76–1.993); 0.399
Age	1.001 (0.99–1.012); 0.841	1.009 (0.996–1.023); 0.166	1.011 (0.997–1.025); 0.133	1.008 (0.995–1.02); 0.238
Head and neck cancer	0.857 (0.54–1.362); 0.514	1.109 (0.704–1.748); 0.656	0.983 (0.423–2.285); 0.968	1.084 (0.686–1.712); 0.73
Esophagogastric cancer	1.96 (1.007–3.816); 0.048	1.974 (1.012–3.851); 0.046	0.897 (0.332–2.422); 0.83	1.985 (1.016–3.879); 0.045
Non-tumoral esophagogastric pathology	4.035 (0.362–44.97); 0.257	4.58 (0.408–51.35); 0.217	*	4.486 (0.432–54.67); 0.2
ALS	0.511 (0.278–0.94); 0.031	0.55 (0.29–1.041); 0.066	0.655 (0.17–2.296); 0.509	0.567 (0.3–1.07); 0.08
Neurological diseases	1.068 (0.605–1.885); 0.82	0.829 (0.462–1.485); 0.527	1.061 (0.509–2.21); 0.875	0.848 (0.479–1.502); 0.572
Maxillofacial pathology	2.099 (0.279–14.447); 0.488	1.754 (0.24–12.791); 0.58	*	1.766 (0.243–12.86); 0.574

* There were no patients in this group.

## Data Availability

The data presented in this study are available on request from the corresponding author.

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
