# Peer review of "Percutaneous Radiology Gastrostomy (PRG)-Associated Complications at a Tertiary Hospital over the Last 25 Years"

_nutrients, 2022, doi:10.3390/nu14224838_

Round 1
Reviewer 1 Report
Thank you for the opportunity to comment on this document. The topic is interesting but after reading it, I have some concerns.
1º I would like to know what is the novelty of the article and its contribution to the knowledge of the scientific community.
2º I do not find this research of special relevance for our journal.
3º The English I think could be improved throughout the text
Author Response
Thank you for the opportunity to comment on this document. The topic is interesting but after reading it, I have some concerns.
1º I would like to know what is the novelty of the article and its contribution to the knowledge of the scientific community.
Dear reviewer, first of all we would like to thank you for taking the time to review our manuscript. We understand your concerns about it.
We believe that our work is novel and differs from what has been published so far for several reasons. The first one is the large number of patients included in it, as it is one of the studies to date with the largest number of patients with PRG from a single center. The fact that they are from a single center we believe is a strength of our study, since "all the PRGs were indicated and followed by the same group of professionals from the Nutrition Unit and interventional radiology, logically with new incorporations and absences during 25 years, but under the same action protocols", as we indicate in the last paragraph of the discussion. Furthermore, published studies on gastrostomy complications tend to focus on pediatric patients or on specific pathologies. By including patients with very diverse pathologies, our study is more representative of the complications of the technique itself, not of the intrinsic characteristics of specific pathologies in relation to the procedure. On the other hand, it differs from other published articles not only because of the number of patients, but also because of the large number of years of experience evaluated. The PRG technique was first described in 1981, although it was subsequently implemented in various centers and was not sufficiently widespread in the world until a few years later. Our article represents all the experience of our center with PRG, so it can be seen how its indication has been increasing and how the pathologies for which it was indicated have been changing. In addition, it can be seen how techniques have been replaced in order to improve results and how complications have been effectively reduced (except with the balloon technique without pexy for the reasons explained in the article). We believe that this representation of the experience of our center (a national and European reference in both Clinical Nutrition and interventional radiology), distinguishes our article from others published in the scientific literature, which are more focused on specific periods of times than on recounting their experience over a long period of daily clinical practice. In the following answer we will also emphasize how our results can be of interest to colleagues in our and other specialties.
2º I do not find this research of special relevance for our journal.
We understand your concerns regarding our article, since it is the first article to be published in Nutrients on complications associated with gastrostomies. However, the aims of this journal are to publish articles "that provide novel insights into the impacts of nutrition on human health" (https://www.mdpi.com/journal/nutrients/about). Gastrostomies are a common enteral nutrition modality in the clinical practice of those who specialize in Clinical Nutrition. In our center, more than 1,100 gastrostomies have been performed in the last 25 years and according to the study by Kohli et al.1 in the United States 184,068 gastrostomies were performed between 2016 and 2017 (154,007 of them with radiological technique). Although gastrostomies are performed by endoscopists, radiologists and surgeons, all of them are indicated by physicians who are dedicated to Clinical Nutrition, and also the follow-up is performed exclusively in these units. Sharing the experience of our center, large in absolute numbers and in time, can be very useful for fellow experts in Clinical Nutrition (as well as other subjects such as endoscopy, interventional radiology, abdominal surgery, oncology, neurology...) from smaller and less experienced centers. The publication of data on the safety of these techniques, such as ours, could support their implementation in other centers where they are not currently available. This is especially relevant in PRG, which are less widespread than endoscopic gastrostomies, as we have added in paragraph 3 of the discussion in the new version of the manuscript:
“… Therefore, we believe that it is important that Clinical Nutrition units with long experience in PRG, such as ours, publish data on complications, as this may encourage other centers to implement or increase the availability of this technique. This could benefit patients, since it would constitute an alternative to PEG in those cases in which it could not be performed and would avoid the performance of surgical techniques in patients in whom PRG would be feasible.”
For all these reasons, we believe that our article is relevant to Nutrients, since it is of interest to specialists in Clinical Nutrition (important readers of this journal); and because the complications associated with these nutrition techniques, which are so frequent in daily clinical practice, have an impact on human health.
1 Kohli, D.R.; Kennedy, K.F.; Desai, M.; Sharma, P. Safety of Endoscopic Gastrostomy Tube Placement Compared with Radiologic or Surgical Gastrostomy: Nationwide Inpatient Assessment. Gastrointest. Endosc. 2021, 93, 1077-1085.e1, doi:10.1016/j.gie.2020.09.012.
3º The English I think could be improved throughout the text
Revised. Our new version of the manuscript has been checked by a native English-speaking colleague.

Reviewer 2 Report
Interesting job overall.
Some tips for the results, avoid repetitions between the results written and the tables.
Tables format should be improved with a cleaner style.
Author Response
Interesting job overall.
Dear reviewer, first of all we would like to thank you for the time you have taken to review our manuscript, as well as the comments you have made that we believe have helped to improve our work.
Some tips for the results, avoid repetitions between the results written and the tables.
Revised. We have removed from the text the data already in the tables and reordered the paragraphs on specific cases of complications associated with the different techniques (now after Table 3) to make the style cleaner and without repetition.
Tables format should be improved with a cleaner style.
Revised. We have adapted the tables to the format of the journal. We believe it has a much cleaner style.

Reviewer 3 Report
This is a correct work. Large experience. Authors should discuss on endoscopic gastrostomy (PEG) and compare both procedures. A stratified analysis of the pts with PRG with different techniques i.e. pigtail or balloon should be analyzed in more details.
Author Response
This is a correct work. Large experience.
Dear reviewer, we would like to thank you for your kind comments about our work, as well as the time spent reviewing it. We would also like to thank you for providing constructive criticism that has helped to improve the original version of the manuscript.
Authors should discuss on endoscopic gastrostomy (PEG) and compare both procedures.
We have added a paragraph in the discussion (now paragraph 3 of this section) in which we compare PRG and PEG, providing several new bibliographic references comparing both techniques in terms of complications and other aspects such as cost. Thanks to this we have also added at the end of the paragraph a few sentences on why we think our study is relevant for the scientific community, especially for specialists in Clinical Nutrition.
A stratified analysis of the pts with PRG with different techniques i.e. pigtail or balloon should be analyzed in more details.
In order to avoid confounding factors, a stratification by specific PRG technique was added by performing a multivariate analysis with each risk factor studied, in addition to the univariate analysis already found in the previous version of the manuscript. These results have been added to Table 4 and a reference to this new analysis has been made in the Methods section, subsection "statistical analysis". We would like to thank you again for the suggested changes as we believe they have improved the quality of our study.

Round 2
Reviewer 1 Report
There is no doubt that the presentation and form of the article has improved, the requests of the reviewers have been met, but I still do not know what is the novelty of the contribution of the authorsin their subject.